# Sedimentary ancient DNA as part of a multimethod paleoparasitology approach reveals temporal trends in human parasitic burden in the Roman period

Marissa L. Ledger[1,2*¤], Tyler J. Murchie[2,3], Zachery Dickson[4], Melanie Kuch[2], Scott D. Haddow[5], Christopher J. Knüsel[6], Gil J. Stein[7], Mike Parker Pearson[8], Rachel Ballantyne[9], Mark Knight[10], Koen Deforce[11,12], Maureen Carroll[13], Candace Rice[14], Tyler Franconi[14], Nataša Šarkić[15], Saša Redžić[16], Erica Rowan[17], Nicholas Cahill[18], Jeroen Poblome[19], Maria de Fátima Palma[20], Helmut Brückner[21], Piers D. Mitchell[1], Hendrik Poinar[2,22,23]

1 Department of Archaeology, The Henry Wellcome Building, University of Cambridge, Cambridge, United Kingdom, 2 McMaster Ancient DNA Centre, Department of Anthropology, McMaster University, Hamilton, Ontario, Canada, 3 Hakai Institute, Heriot Bay, British Columbia, Canada, 4 Department of Biology, McMaster University, Hamilton, Ontario, Canada, 5 Department of Cross-Cultural and Regional Studies, University of Copenhagen, Copenhagen, Denmark, 6 CNRS, MC, PACEA, UMR, Université de Bordeaux, Pessac, France, 7 Institute for the Study of Ancient Cultures (ISAC), University of Chicago, Chicago, Illinois, United States of America, 8 Institute of Archaeology, University College London, London, United Kingdom, 9 McDonald Institute for Archaeological Research, University of Cambridge, Cambridge, United Kingdom, 10 Cambridge Archaeological Unit, Department of Archaeology, University of Cambridge, Cambridge, United Kingdom, 11 Department of Archaeology, Ghent University, Ghent, Belgium, 12 Royal Belgian Institute of Natural Sciences, Brussels, Belgium, 13 Department of Archaeology, University of New York, New York, United Kingdom, 14 Joukowsky Institute for Archaeology & the Ancient World and Department of Classics, Brown University, Providence, Rhode Island, United States of America, 15 Aita Bioarchaeology, Barcelona, Spain, 16 Institute of Archaeology, Belgrade, Serbia, 17 Department of Classics, Royal Holloway, University of London, London, United Kingdom, 18 Department of Art History, University of Wisconsin–Madison, Madison, Wisconsin, United States of America, 19 Unit of Archaeology, KU Leuven, Belgium, 20 Campo Arqueológico de Mértola/CEAACP, Mértola, Portugal, 21 Institute of Geography, University of Cologne, Cologne (Köln), Germany, 22 Department of Biochemistry and Biomedical Sciences, McMaster University, Hamilton, Ontario, Canada, 23 Michael G. DeGroote Institute for Infectious Disease Research, McMaster University, Hamilton, Ontario, Canada

¤ Current Address: Department of Pathology and Molecular Medicine, McMaster University, Hamilton, Ontario, Canada.
* ledgerm@mcmaster.ca

## Abstract

The detection of parasite infections in past populations has classically relied on microscopic analysis of sediment samples and coprolites. In recent years, additional methods have been integrated into paleoparasitology such as enzyme-linked immuno-sorbent assay (ELISA) and ancient DNA (aDNA). The aim of this study was to evaluate a multimethod approach for paleoparasitology using microscopy, ELISA, and sedimentary ancient DNA (sedaDNA) with a parasite-specific targeted capture approach and high-throughput sequencing. Using 26 samples dating from c. 6400 BCE to 1500 CE that were previously analyzed with microscopy and ELISA, we aimed to more accurately detect and reconstruct parasite diversity in the Roman Empire and compare this

the Creative Commons Attribution License, which permits unrestricted use, distribution, and reproduction in any medium, provided the original author and source are credited.

**Data availability statement:** The data that support the findings of this study are publicly available from the NCBI Sequence Read Archive with the accession PRJNA1194279.

**Funding:** This work was supported by the Social Sciences and Humanities Research Council of Canada (752-2016-2085 to MLL), a Tidmarsh Cambridge Scholarship from the Cambridge Commonwealth, European and International Trust and Trinity Hall College, and the Society for the Promotion of Roman Studies (to MLL), a CANA Foundation grant (to TJM and HNP), the Garfield Weston Foundation (to TJM), the Social Sciences and Humanities Research Council of Canada (767-2016-2288 to TJM), the Canadian Institute for Advanced Research, and the Natural Sciences and Engineering Research Council of Canada (to HP). The funders had no role in study design, data collection and analysis, decision to publish, or preparation of the manuscript.

**Competing interests:** The authors have declared that no competing interests exist.

diversity to earlier and later time periods to explore temporal changes in parasite diversity. Microscopy was found to be the most effective technique for identifying the eggs of helminths, with 8 taxa identified. ELISA was the most sensitive for detecting protozoa that cause diarrhea (notably *Giardia duodenalis*). Parasite DNA was recovered from 9 samples, with no parasite DNA recovered from any pre-Roman sites. Sedimentary DNA analysis identified whipworm at a site where only roundworm was visible on microscopy, and also revealed that the whipworm eggs at another site came from two different species (*Trichuris trichiura* and *Trichuris muris*). Our results show that a multimethod approach provides the most comprehensive reconstruction of parasite diversity in past populations. In the pre-Roman period, taxonomic diversity included a mixed spectrum of zoonotic parasites, together with whipworm, which is spread by ineffective sanitation. We see a marked change during the Roman and medieval periods with an increasing dominance of parasites transmitted by ineffective sanitation, especially roundworm, whipworm and protozoa that cause diarrheal illness.

## Author summary

A multimethod approach in paleoparasitology, combining microscopy, ELISA and aDNA, provides a more complete reconstruction of past parasite diversity. Applying this multimethod approach to a diverse set of archeological sediment samples from 6400 BCE to 1500 CE provided insights into temporal changes in parasite diversity in the human past. Microscopy can be used as an effective screening tool for helminths in paleofecal samples, while ELISA is necessary for detection of protozoa, and sedaDNA using targeted enrichment can identify additional taxa and confirm species identification. Targeted enrichment using a comprehensive parasite bait set allows for detection of ancient human parasites and recovery of ancient parasite DNA from as little as 0.25 g of sediment. This multimethod approach reveals that parasite diversity decreased in the Roman period due to a decrease in zoonotic parasites and concurrent increase in fecal-oral parasites, a pattern which is consistent during the medieval period.

## Introduction

Ancient DNA (aDNA) has increasingly been used to improve the detection and characterization of ancient human pathogens [1–4]. Paleogenetic studies of ancient human pathogens have aided in the determination of the causative agents of past epidemics and endemic diseases [3,5,6], identified extinct strains of bacteria and viruses [7,8], and reconstructed ancient genomes of these organisms, allowing for the calibration of mutation rates and investigation of evolutionary histories such as changes in virulence genes [9,10]. However, many infectious diseases, including most parasites, cannot be detected using skeletal remains which are commonly used

in aDNA studies. Parasites contributed to a significant burden of disease in the past, however our ability to detect and study their evolutionary history has been hampered by difficulty accessing their DNA. Paleofeces (preserved fecal material) is the best source for recovering preserved DNA from enteric pathogens including bacteria [11,12], parasites [13–15], and viruses [16,17]. The field of paleoparasitology has a long history of using sediment samples to study enteric parasites in past populations [18]. Samples used in paleoparasitology include coprolites (mineralized feces), pelvic sediment from burials where the intestines decomposed, and sediment containing high amounts of human fecal material such as the fill of latrines and sewer drains. Using these samples to recover genetic material and human gastrointestinal pathogens opens another route for understanding past human health, disease, and lifeways, and provides insight into a large number of pathogens that cannot be diagnosed using skeletal remains.

The use of genetic methods to detect human-infecting parasites in both clinical and ancient samples has lagged behind most other pathogens [19,20]. The main methods used in paleoparasitology are microscopy, and to a smaller extent, enzyme-linked immunosorbent assay (ELISA, an immunological method that can detect antigens from specific organisms of interest). In recent years, the number of studies using aDNA to detect human-infecting parasites has increased (see [20] for a review; and subsequent studies [12,13,15,21–30]). However, aDNA analysis contributes only a small minority of the total investigations performed within paleoparasitology and many studies still rely on a small number of PCR targets.

Here we present a multimethod approach for recovery of ancient parasites using sediment samples from contexts known to contain fecal material, including latrines, sewers, soil from the pelvic area of skeletons, and coprolites. Our multimethod approach incorporates sedaDNA methods using a targeted enrichment approach for parasite-specific DNA, in conjunction with microscopy and ELISA. The aim of this multimethod approach was to generate a more complete understanding of parasite diversity in Europe and the Eastern Mediterranean during the Roman Empire as compared to pre- and post-Roman periods. We analyzed paleofecal samples originating from archeological sites temporally spanning the Neolithic through the medieval period. We used sedaDNA extraction methods previously shown to increase aDNA recovery by 7–20 fold compared to commercial kits [31], paired with targeted enrichment in order to preferentially sequence parasite DNA of interest and avoid the high sequencing costs associated with deep shotgun sequencing for low abundance targets such as pathogenic organisms. This sedaDNA approach was originally designed for paleoecological reconstructions from permafrost [31]—here we show that this approach can also be used to recover the aDNA of human parasites from archeological sediments. Similarly, we show that a multimethod approach to paleoparasitology results in a more comprehensive taxonomic recovery compared to only utilizing a single method. Applying this multimethod approach to paleofecal samples from the Neolithic through medieval periods reveals trends in parasite infection that demonstrate marked change in the dominant species during the Roman period.

## Methods

We analyzed archeological sediments from latrine fill, drain fill, coprolites, and soil from the pelvic area of skeletons. All samples came from human contexts in archeological sites ranging from the Neolithic period (ca. 6400 BCE) through the medieval period (ca. 1500 CE) in regions that were at one time part of the Roman Empire (see Fig 1). A total of 14 sites were included in the study; four from the pre-Roman period, seven from the Roman period, and three from the post-Roman or medieval period. From some sites multiple samples were studied if available, thus a total of 26 samples were included . Further information about each site and samples analyzed can be found in S1 Appendix.

All samples were first studied and reported on using microscopy and ELISA following standard methods [32,33]. The samples selected for this study had previously yielded parasite eggs or came from sites where eggs were found in contextually associated samples [33–43]. For microscopy, a 0.2 g subsample was disaggregated in 0.5% trisodium phosphate. This was then microsieved to collect material between 20 and 160 µm. This fraction of the subsample was mixed with glycerol and viewed under a light microscope (Olympus BX40F) at 200x and 400x magnification to identify preserved helminth eggs based on morphological characteristics.

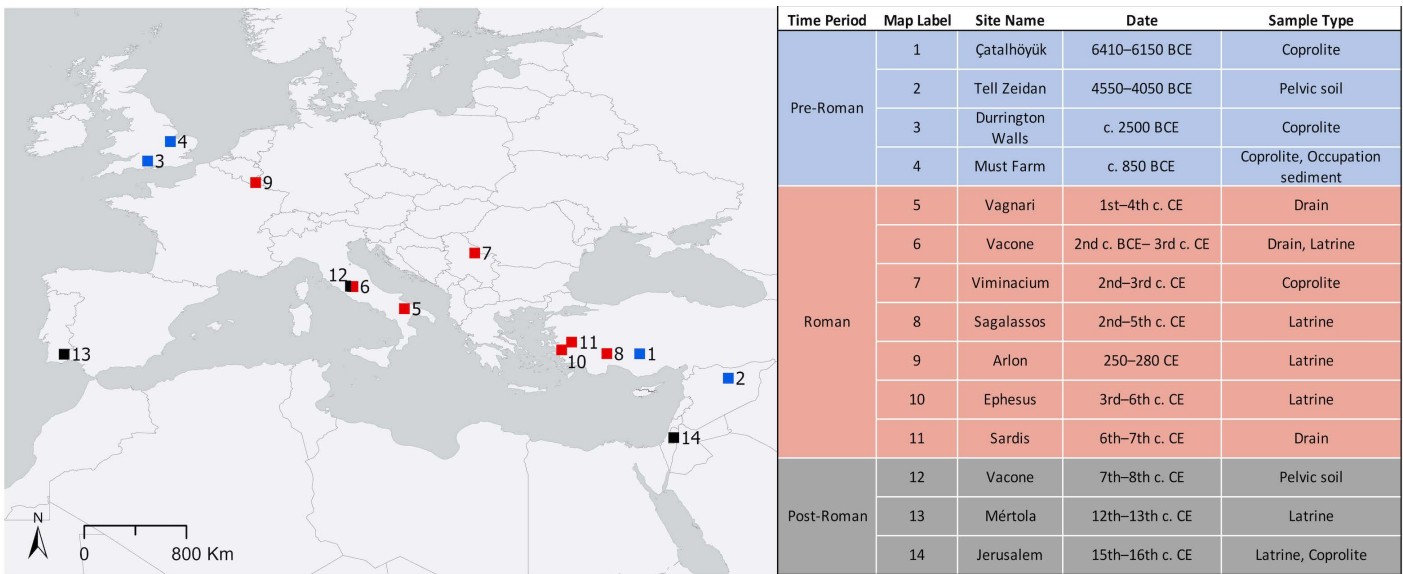

**Fig 1. Map of archeological sites studied.** Map indicates the location of sediment samples studied. The table on the right provides the site name, date of samples, and sample types. Basemap from Natural Earth Basemap (https://www.naturalearthdata.com).

For ELISA, a 1 g subsample was disaggregated in 0.5% trisodium phosphate and microsieved. Given the smaller size of protozoan cysts (less than 20 μm), the material in the catchment container below the 20 μm sieve was collected and concentrated to be used in commercial ELISA kits following the manufacturer's protocols. The kits used were *GIARDIA II*, *E. HISTOLYTICA II*, and *CRYPTOSPORIDIUM II* from TECHLAB, Inc (Blacksburg, USA). These kits are designed for detection of *Giardia duodenalis*, *Entamoeba histolytica*, and *Cryptosporidium* spp. in modern human fecal samples. These kits and those from other manufacturers have been used to detect protozoan antigens in ancient human fecal samples in many previous studies [33,44–47].

For sedaDNA, all work was done in dedicated ancient DNA facilities. No modern molecular work is performed in these labs. A unidirectional workflow is followed with work starting in the reagent cleanroom and proceeding through extraction rooms followed by amplification rooms. Standard precautions to prevent lab and sample contamination in aDNA were followed including wearing a full suit, gloves, and mask. All surfaces are regularly cleaned with 6% sodium hypochlorite and hoods are exposed to UV radiation. The sedaDNA methods used are described below, for further details see S1 Appendix.

First, 0.25 g of material was subsampled. Organic and inorganic material was chemically and physically disintegrated to release DNA using a lysis buffer (see S1 Appendix for reagents and concentrations) placed with the subsample in garnet PowerBead tubes (Qiagen) already containing 750 μL of 181 mM $NaPO_4$ and 121 mM guanidinium isothiocyanate with garnet beads for physical disruption [31]. The samples were vortexed for 15 minutes to mechanically break down the organo-mineralized content and parasite eggs. Bead beating has been shown in clinical and archeological studies to improve DNA recovery by breaking parasite eggs [48,49]. Proteinase K was added after bead beating, and tubes were continuously rotated in an oven set to 35°C overnight. The supernatant was removed and mixed with high-volume Dabney binding buffer as per published protocols [50]. Samples were then centrifuged at 4500 rpm at 4°C following Murchie et al. [31] for a minimum of 6 hours to remove inhibitors; if supernatant was clear at this point, the sample was removed from the centrifuge, and if not, it was centrifuged further for a total of 24 hours. Centrifugation at refrigerated temperatures has been shown to increase the recovery of sedaDNA from complex environmental sample types through the precipitation of enzymatic inhibitory compounds commonly found in sediment and fecal samples [31]. The remainder of the extraction and purification followed Dabney et al. [50] with the binding buffer passed through silica columns and eluted in 50 μL elution buffer.

DNA libraries were prepared for Illumina sequencing using a double-stranded method [51,52] with minor modifications for blunt end repair as per our laboratory protocols (see S1 Appendix). A subset of four libraries (MP12, MP26, MP37, and MP53) were indexed separately for shotgun sequencing with a targeted sequencing depth of 2 million reads per sample. Targeted enrichment for parasite DNA was carried out using in-solution hybridization capture with RNA baits designed using modern parasite genomes (see S2 Appendix for reference sequence accession numbers). The parasite bait set contained 30,240–75mer baits based on mitochondrial and nuclear gene sequences retrieved from GenBank NCBI [53,54] from a list of identified human and animal infecting parasites and protozoa. Tiled baits were designed to capture 121 parasite taxa (see S2 Appendix). A progressive tiling approach was used to balance the number of baits for each taxa as best as possible. With this approach the bait design algorithm determined the maximum and minimum tiling density possible for each bait region. Taxa with few bait regions were maximally tiled to increase bait number while taxa with many potential bait regions were minimally tiled to balance bait number per taxa. The final average tiling density per taxa ranged from 2x to 30x. The number of baits for each taxa was ultimately limited by reference sequences available and ranged from 2–922 baits per taxa. Baits were purchased from Daicel Arbor Biosciences and our enrichment protocol followed the MYbaits V4 protocol supplied by the manufacturer.

Libraries for shotgun sequencing and targeted enrichment were quantified using qPCR and pooled to equimolar concentrations based on the molarity of each library. Pools were size-selected using electrophoresis followed by gel excision using the QIAquick Gel Extraction Kit (Qiagen). Paired-end sequencing was conducted using 2x90 sequencing chemistry on an Illumina HiSeq 1500 platform at the McMaster Farncombe Metagenomics Facility.

Bioinformatic workflow followed previously described in-house protocols for sedaDNA analysis (see [31]). In brief, *leeHom* [55] was used for trimming and merging, reads were then mapped using *network-aware-BWA* to the parasite reference sequences that had been used to design the bait set. These mapped reads were size filtered to those greater than or equal to 24 bp and string de-duplicated with *NGSeXplore*. Filtered and mapped reads were run through BLASTn to return the top 100 taxonomic hits. MEGAN was then used to assign these reads to the lowest common taxonomic rank. For further detail on bioinformatic workflow see S1 Appendix. Raw sequence data has been deposited in the NCBI Sequence Read Archive (PRJNA1194279).

## Results

### Pre-Roman periods

Microscopic analyses of all samples described here has been previously published. *Trichuris trichiura* (whipworm) was found in the human coprolite from 6410–6150 BCE Çatalhöyük, Türkiye [38]. *Schistosoma* sp. was found in pelvic soil from a burial (5800–4000 BCE) in Tell Zeidan, Syria [34]. One coprolite from the Neolithic settlement (2500 BCE) of Durrington walls near Stonehenge, Britain, was included which contained Capillariid eggs [43]. This coprolite was previously identified as human in origin using fecal lipid biomarkers. Interestingly, DNA sequenced from this coprolite contained a high proportion of reads assigned to *Canis lupus familiaris* with very few reads assigned to *Homo sapiens* (see S1 Appendix). At the Bronze Age site of Must Farm in Britain (c. 850 BCE), sediment samples and human and dog coprolites were included. The source of these coprolites had also been previously identified using fecal lipid biomarkers [39]. One of the coprolites previously identified as human and one of dog origin underwent shotgun sequencing and were shown to contain high amounts of DNA from *Canis lupus familiaris* (see S1 Appendix). Microscopy of coprolites found eggs of *Capillaria* sp., *Echinostoma* sp., fish tapeworm (*Dibothriocephalus* sp.), giant kidney worm (*Dioctophyma renale*), and whipworm (*Trichuris* sp.). Occupation layer sediment contained all of these taxa except *Echinostoma* sp. [39].

The only Pre-Roman samples studied using ELISA were those from Must Farm as there was insufficient remaining material from the other samples. The ELISA results from these samples were indeterminate as all sample wells (8 technical replicates on two separate biological replicates for each sample) were positive for all three species (*Giardia duodenalis*, *Entamoeba histolytica*, and *Cryptosporidium* spp.). This is an exceptionally unusual result for paleoparasitological studies and suggests a false positive result likely from cross-reactivity with another biomolecule within the samples. Furthermore, *Cryptosporidium* spp. have never been detected in ancient samples from Europe.

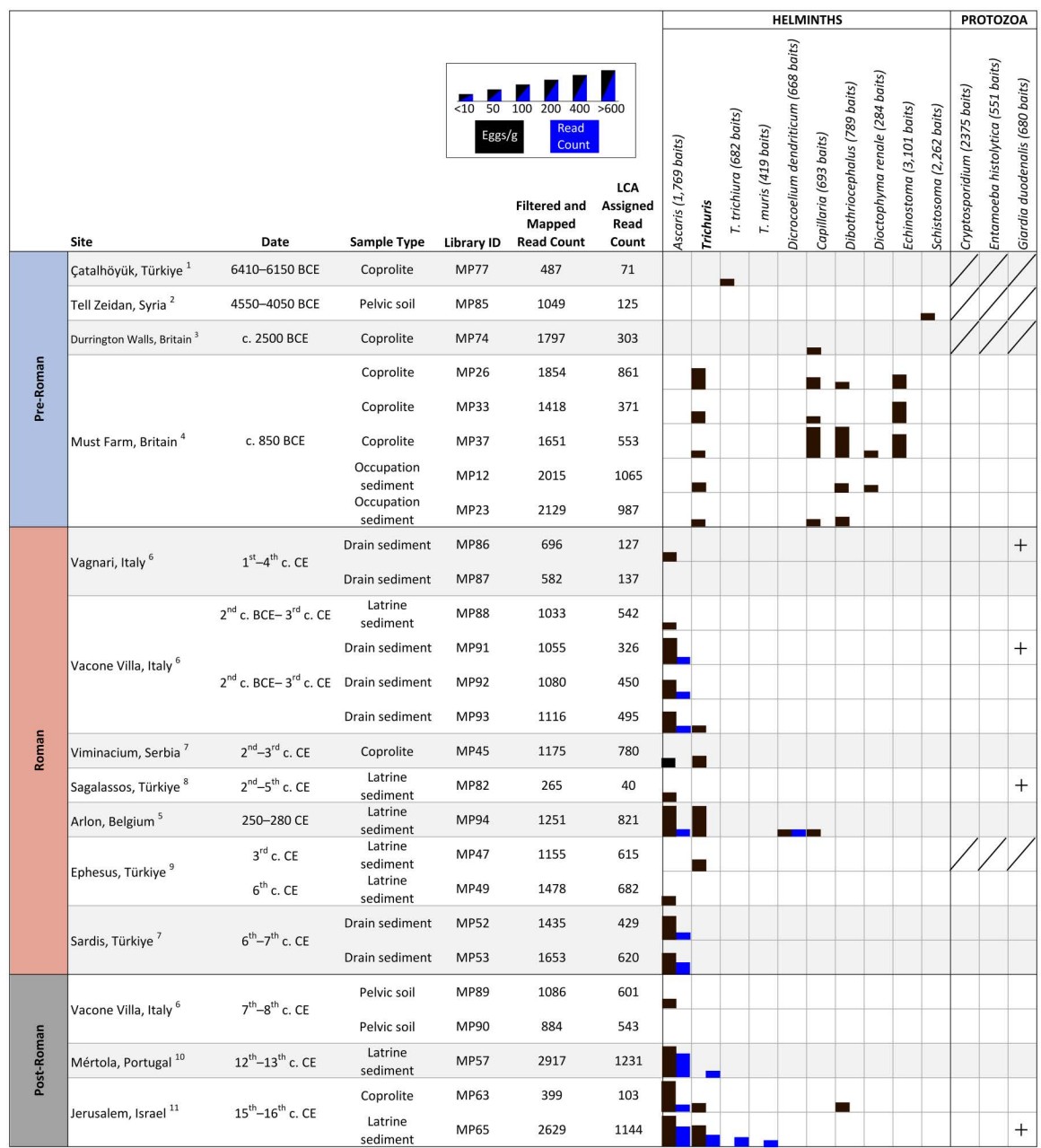

**Fig 2. Summary of parasites recovered from each sample using microscopy, ELISA, and ancient DNA.** Black bars represent egg concentrations identified using microscopy and blue bars represent DNA read count. Samples for which insufficient material was available for ELISA are crossed out and positive ELISA tests are marked with a +. 1 Ledger et al., 2019 [38]; 2 Anastasiou et al., 2014 [34]; 3 Mitchell et al., 2022 [43]; 4 Ledger et al., 2019 [39]; 5 Deforce et al., 2021 [42]; 6 Ledger et al., 2021 [33]; 7 Ledger et al., 2020 [41]; 8 Williams et al., 2017 [36]; 9 Ledger et al., 2018 [37]; 10 Knorr et al., 2019 [40]; 11 Yeh et al., 2015 [35].

No parasite DNA was recovered from the 8 Pre-Roman samples included (Fig 2). Extracts were quantified using qPCR after library preparation using probes designed for the ligated sequencing adapters. The quantification of adapter-ligated molecules in these libraries (ranging from $10^6$ to $10^9$ copies/uL) was consistent with other samples where parasite DNA was recovered (see Table Di in S1 Appendix). Additionally, the number of filtered reads mapped to the parasite reference

sequences and subsequently LCA assigned were consistent with samples from the Roman and post-Roman periods (Fig 2). Within the pre-Roman samples, 59–93% of the LCA-assigned reads were assigned to bacteria, while less than 10% of reads were assigned to Fungi, Metazoa, Sar, and Viridiplantae with the exception of one sample where 15% of reads were assigned to Fungi. Though reads assigned to Metazoa and Sar may be from parasites, these reads could not be assigned below the Clade level (e.g., Stramenopiles, Alveolata, Bilateria) thus are relatively uninformative. Thus, the majority of LCA-assigned reads in the pre-Roman samples were from bacteria and we were unable to recover parasite DNA from taxa of which eggs were preserved.

## Roman

In the 13 Roman period samples, the parasite taxa recovered by microscopy were less diverse than those of pre-Roman sites. Roundworm (*Ascaris* sp.) was recovered from 11 samples studied, whipworm eggs were found in four, and both *Capillaria* sp. and *Dicrocoelium dendriticum* in one.

The only protozoa recovered from Roman period sites using ELISA was *Giardia duodenalis*. *Giardia duodenalis* was detected in drain samples from Vagnari and the Villa of Vacone in Italy, and in a latrine from Sagalassos in Türkiye [33].

For aDNA analysis, reads assigned to roundworm (*Ascaris* sp.) were recovered from six samples (Fig 2). *Ascaris* sp. eggs were found by microscopy in all of these samples. *Dicrocoelium dendriticum* DNA was also recovered from the latrine at Arlon, Belgium; *Dicrocoelium* sp. eggs were also found by microscopy from this site. No protozoa DNA was recovered.

## Post-Roman

Similar to the Roman period, the most common helminth recovered from medieval period sites using microscopic analysis was roundworm (*Ascaris* sp.) which was found in four of five samples. Whipworm was found in two samples and fish tapeworm in one sample. The only protozoa recovered using ELISA was *Giardia duodenalis*. This was found in the latrine from Jerusalem [35].

Ancient DNA from roundworm (*Ascaris* sp.) was found in three medieval samples studied (Fig 2). In the latrine sediment from Mértola, Portugal three of these reads were assigned to the species-level as *Ascaris lumbricoides*, the human infecting species of *Ascaris*, using MEGAN. However, when they were mapped to the reference genomes of both *Ascaris lumbricoides* (GCA_015227635.1) and *Ascaris suum* (GCA_013433145.1) these reads had 100% identity with both species as would be expected based on the limited genetic diversity between these species (see S1 Appendix). In addition, reads assigned to whipworm (*Trichuris* sp.) were found in the latrine from Mértola in Portugal. Interestingly, no *Trichuris* sp. eggs were seen on microscopic analysis of this latrine sediment. Species-level identification was possible for whipworm reads from the latrine in Jerusalem. Reads were not only assigned to human whipworm (*Trichuris trichiura*), but also mouse whipworm (*Trichuris muris*). These results are comparable to results of a recent aDNA study employing shotgun sequencing to characterize microbial and eukaryotic DNA in a separate subsample from this same latrine in Jerusalem. In that study, *Ascaris lumbricoides* and *Trichuris trichiura* were detected [12].

## Expansion of results using ancient DNA

Shotgun sequencing on a subset of four samples was unable to recover any identifiable parasite DNA. However, using a targeted enrichment approach, parasite DNA was recovered from 9 out of 26 samples (see Fig 2 for summarized results and S3 Appendix for results by individual sample). The majority of sequenced reads assigned to human parasites were from roundworm (*Ascaris* sp.). Additional parasite taxa from which we were able to recover DNA included the lancet liver fluke (*Dicrocoelium dendriticum*) and whipworm (*Trichuris* spp.) (Fig 2). In one of these samples (MP57 from Mértola, Portugal) whipworm eggs were not visualized by microscopy thus, aDNA analysis provides evidence for additional taxa.

The microscopic identification of *Ascaris* sp. was replicated using aDNA analysis in six out of a total 11 Roman period samples from which *Ascaris* sp. eggs were identified by microscopy. Three out of five medieval period samples in which *Ascaris* sp. eggs were identified by microscopy contained *Ascaris* sp. DNA. From both of these time periods, only three of these samples contained *Ascaris* sp. eggs that were sufficiently preserved to still have a visible outer mamillated coat. Mamillated *Ascaris* sp. eggs were recovered from Arlon in Belgium, Viminacium in Serbia, and the Mamluk period cesspit in Jerusalem. The mamillated coat of *Ascaris* spp. is an important characteristic feature used for identification of the eggs, however, it can be absent in modern eggs and is often lost in archeological contexts which makes identification challenging [56–59]. *Ascaris* sp. DNA was detected in samples containing decorticated and poorly preserved *Ascaris* sp. eggs as can be seen in Fig 3.

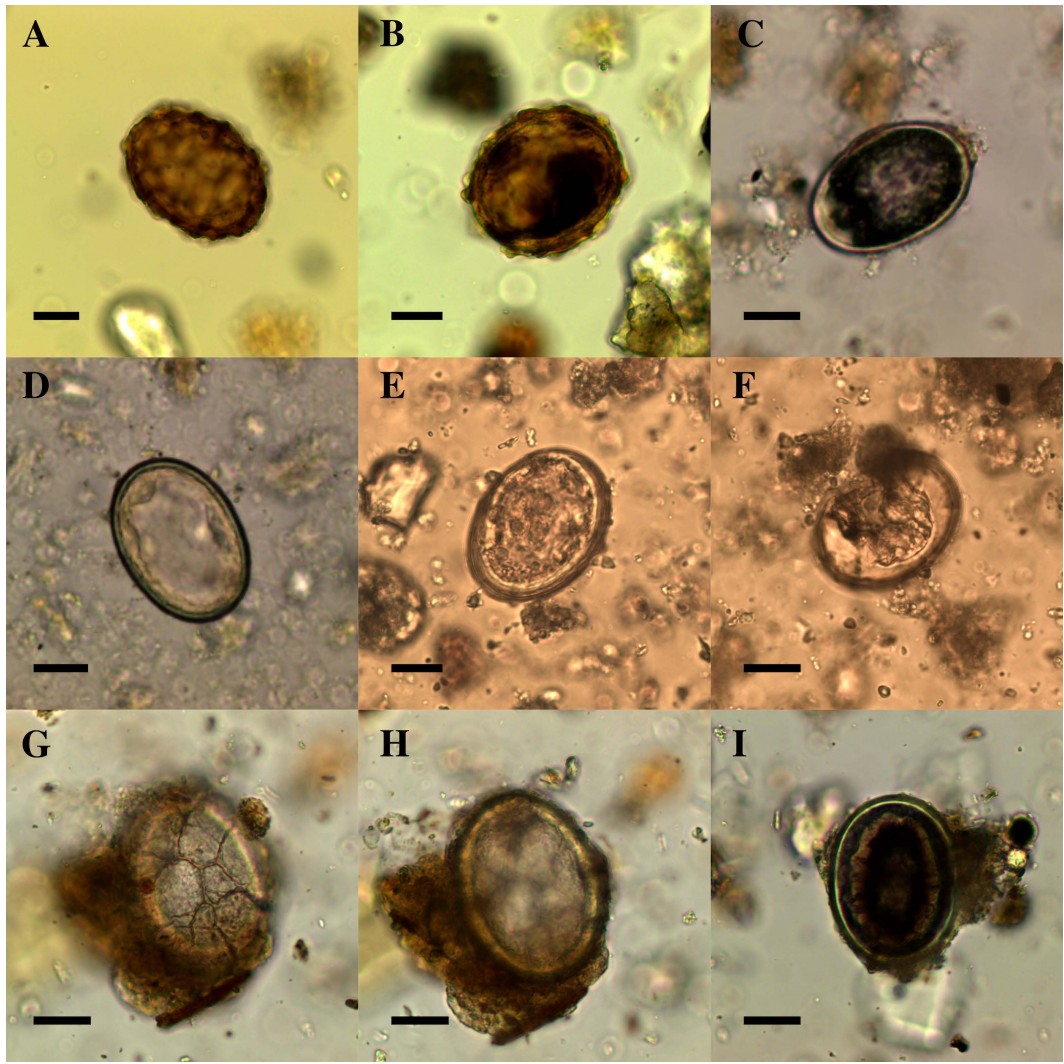

**Fig 3. Variation in morphological appearance of *Ascaris* sp. eggs identified in samples with confirmed presence of *Ascaris* sp. using aDNA methods.** (A) fertilized egg from Arlon with mamillated coat; (B) fertilized egg from Arlon with mamillated coat; (C) fertilized decorticated egg from Mértola (image credit: Delaney Knorr); (D) fertilized decorticated egg from Mértola (image credit: Delaney Knorr); (E) fertilized decorticated egg from Vacone; (F) fertilized decorticated egg from Vacone; (G and H) fertilized decorticated egg from Sardis with surface cracking; (I) fertilized decorticated egg from Sardis. Scale bars indicate 20 μm.

| Site | Sample Type | DNA Library ID | *Ascaris* (eggs per gram) | % LCA Assigned (Ascaridoidea or Lower) | Read Count (Ascaridoidea or Lower) |
|---|---|---|---|---|---|
| Jerusalem | Cesspit sediment | MP65 | 1580 | 21.8% | 249 |
| Mértola | Latrine | MP57 | 1110 | 24.1% | 297 |
| Arlon | Latrine | MP94 | 990 | 0.5% | 4 |
| Jerusalem | Coprolite | MP63 | 595 | 5.8% | 6 |
| Vacone | Drain sediment | MP91 | 405 | 1.8% | 6 |
| Sardis | Drain sediment | MP52 | 305 | 1.2% | 5 |
| Sardis | Drain sediment | MP53 | 205 | 9.7% | 60 |
| Vacone | Drain sediment | MP93 | 185 | 1.4% | 7 |
| Vacone | Drain sediment | MP92 | 140 | 1.6% | 7 |
| Vacone | Pelvic soil | MP89 | 30 | 0.0% | 0 |
| Vagnari | Drain sediment | MP86 | 25 | 2.4% | 3 |
| Ephesus | Latrine | MP49 | 25 | 0.0% | 0 |
| Sagalassos | Latrine | MP82 | 20 | 0.0% | 0 |
| Vacone | Latrine | MP88 | 5 | 0.0% | 0 |

**Fig 4. Heatmap comparing concentration of *Ascaris* sp. eggs recovered from each sample to percentage of LCA-assigned reads.** *Ascaris* sp. egg concentration is reported in eggs per gram based on microscopy and LCA-assigned reads include reads that were assigned to the Superfamily Ascaridoidea or lower and read count assigned to Ascaridoidea or lower. Samples are ordered in the table based on egg per gram concentration from highest to lowest concentration with dark red background for the lowest concentrations, yellow/orange mid-range and green the highest concentrations.

Fig 4 contains a comparison between the concentration of roundworm (*Ascaris* sp.) eggs recovered by microscopy to the percent of LCA-assigned reads to Ascaridoidea or lower taxonomic rank and read count of the same. Unsurprisingly, the samples that contained the highest concentration of *Ascaris* sp. eggs also had a higher percentage of reads assigned to *Ascaris* sp. For nearly all samples where *Ascaris* sp. eggs were identified by microscopy but no DNA was recovered, the egg concentration was 30 eggs per gram or less.

The sequence reads assigned to roundworm were authenticated as aDNA using *mapDamage* [60]. They showed typical deamination patterns for aDNA and fragment length distributions (Fig 5). Statistical analysis could not be done on reads from other taxa due to the low number of reads.

## Discussion

### A multimethod approach in paleoparasitology

Each individual method used in our study contributed complementary results to the reconstruction of parasite diversity in the past. For samples from the Neolithic and Bronze Age (where the number of surviving parasite eggs was generally low), microscopy was more sensitive for detecting ancient parasites than was aDNA. As can be seen in Fig 2, six different taxa were recovered using microscopy and this included rare parasites in archeological settings such as *Schistosoma* sp., *Echinostoma* sp., and *Dioctophyma renale*. No parasite DNA sequences were recovered in these samples using our parasite sedaDNA methods, despite all of the taxa being present in our bait set (see S2 Appendix for a full list of baits); with 2,262 baits targeting *Schistosoma* spp., 3,101 baits for *Echinostoma* spp., and 284 baits for *Dioctophyma renale*. The majority of DNA recovered from these samples was bacterial in origin. The parasite diversity in these samples was more diverse than that seen in the Roman and medieval periods, and is further characterized by a notable absence of *Ascaris* sp. which is the most common parasite found in later time periods. This raises the possibility that our targeted capture

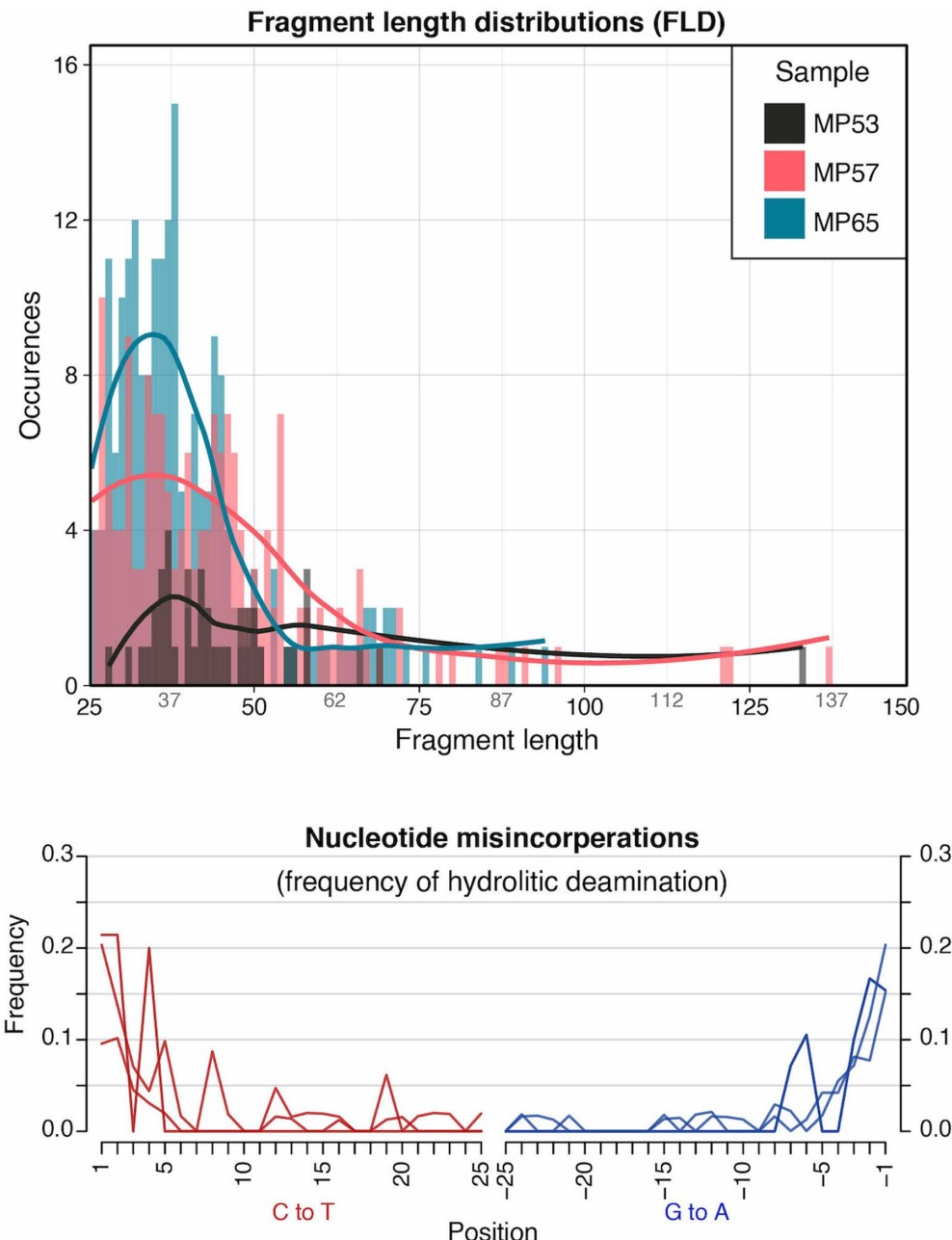

**Fig 5. Deamination and fragment length distributions for *Ascaris* sp. reads.** Data generated using *mapDamage* showing C to T and G to A deamination patterns (top) and fragment length profile (bottom) for reads assigned to *Ascaris* sp. in samples with highest read count.

approach was not as efficient at detecting these parasite species as it was for *Ascaris* spp. Aside from parasite DNA recovery, our sedaDNA methods provided additional data for the identification of the source of coprolites studied. High read counts assigned to dog (*Canis lupus familiaris*) in the Durrington Walls and Must Farm coprolites suggests that these coprolites may in fact be from dogs not humans as previously thought. The accurate identification of coprolites is important for interpretation of parasite results.

Roundworm (*Ascaris* spp.) and whipworm (*Trichuris* spp.) were the most commonly identified parasites by both microscopy and aDNA in the Roman and medieval period samples. In all Roman period samples in which parasite DNA was recovered, eggs from the assigned taxa were identified by microscopy. No additional parasites were detected using ancient DNA analysis for our Roman period sites, and aside from reads assigned to the lancet liver fluke (*Dicrocoelium dendriticum*) in the latrine from Arlon, Belgium, all other reads were from *Ascaris* sp. and could not be assigned to a lower rank than the genus-level. This was not the case for the medieval period samples. In medieval period samples additional taxa were recovered that were not identified using microscopy. The only time period where whipworm (*Trichuris* spp.) DNA was recovered was from our medieval period samples, despite *Trichuris* sp. being identified by microscopy in Roman period samples.

Combining aDNA with microscopy in paleoparasitology provides the potential to identify ancient parasites to the species-level [14,19,21]. For many parasite taxa, egg morphology cannot be used to accurately identify eggs to the species-level [19,61,62]. We were able to do this in one case using our targeted enrichment approach. Our bait design used a hierarchical approach where baits specific to the species-level were included when possible as well as baits for higher taxonomic levels to increase our ability to capture preserved sequences even outside species-level resolution. This design was used to increase our ability to capture parasite DNA while also providing the opportunity to make species-level identification if appropriate sequences were present. Two species of *Trichuris* were identified in the latrine sediment from 15th–16th c. CE Jerusalem. Both the human-infecting species, *Trichuris trichiura*, and the mouse-infecting species, *Trichuris muris*, were identified. The presence of *T. muris* in the latrine could be a result of mice having access to the latrine or indicate that material contaminated with feces from rodents was deposited in the latrine. In latrine sediment from the medieval site of Mértola, *Ascaris* reads were assigned to the species-level as *Ascaris lumbricoides* by MEGAN, however on further interrogation of these reads we find that they are not unique to *Ascaris lumbricoides* but also match equally well to *Ascaris suum* (classically considered to infect pigs). Given extremely little genetic difference between these two species, it is not surprising that for all other samples, reads could only be assigned to the genus-level [63,64]. Attempts at molecular characterization of *A. suum* and *A. lumbricoides* have revealed very low divergence rates in ITS1 and mitochondrial regions (~0.01%) [65,66]. Comparison of full mitochondrial genomes of *A. lumbricoides* and *A. suum* from China showed 98.1% sequence identity [63]. In addition, species-designation has been questioned on the basis of experimental studies showing that the life-cycle of both species can be completed in either host [67–69]. Thus, the studies above and other authors have questioned the species distinction between *A. lumbricoides* and *A. suum* [70].

One important contribution of our study is that aDNA analysis was able to replicate identification of *Ascaris* sp. in samples that contained eggs which had lost their mamillated coat as a result of degradation and environmental conditions after egg deposition. The outer mamillated coat of *Ascaris* spp. eggs is a key morphological feature used in identification of the eggs. The *Ascaris* sp. eggs identified by microscopy in samples from this study were all decorticated except for those from Arlon, Viminacium, and Jerusalem (Fig 3). Thus, if one is only identifying eggs with preserved mamillated coats, they will grossly underestimate the prevalence of *Ascaris* spp. in the past. We cannot be certain from which source our *Ascaris* sp. DNA stems. As DNA was not extracted directly from the eggs, it is possible that *Ascaris* sp. DNA was present in the sediment outside of these decorticated eggs. However, given the correlation of egg concentrations with DNA content in the samples (see Fig 4), it is most likely that this DNA originated from the eggs themselves. Importantly, the overall morphological consistency of these eggs with decorticated *Ascaris* sp. eggs would argue in favor of the origins of these DNA molecules. Even in modern studies, decorticated roundworm eggs can be easily misidentified [59]. Thus, the addition of aDNA analysis can confirm identification in cases of uncertainty due to taphonomic changes and poor egg preservation and allows us to more confidently explore the morphologic appearance of ancient eggs.

The targeted enrichment approach taken allows for the potential recovery of any of the 121 taxa included in the bait set (S2 Appendix), which is the largest targeted approach in paleoparasitology to date without relying on shotgun sequencing. Many paleogenetic studies in parasitology still rely on PCR which has a number of limitations. First, PCR amplicons

often range between 50–600 bp, with most over 100 bp [71–74]. However, most aDNA fragments are smaller than 100 bp, with fragment length distribution modes generally around 40–50 bp [75,76]. Secondly, with targeted enrichment one can include baits designed to capture large genomic regions from mitochondrial and nuclear loci in one reaction. Most studies in paleoparasitology relying on PCR methods targeted a maximum of 5 taxa or even a single taxon previously identified by microscopy [13,26,29,30,48,74]. Targeted enrichment has only previously been used in parasite aDNA research for detecting malaria (*Plasmodium* sp.) in skeletal remains [77]. On the other hand, while we are limited by coverage of the bait set and references sequences available for robust bait design, targeted enrichment allowed for recovery of parasite DNA when shotgun sequencing to an average depth of 3.3 million reads (total read counts from shotgun sequencing ranged from 2.2–4.0 million reads) was unsuccessful in recovering any parasite DNA. In published paleoparasitological studies relying on shotgun sequencing, most samples were sequenced to a depth of at least 5 million reads with some samples being sequenced to a depth of 60 million reads [12,14,21]. The cost of sequencing to this level is not financially feasible for many studies and most aDNA and paleoparasitology labs. Additionally, in most cases less than 5% of these sequenced reads are assigned to target organisms [78]. In our samples even with targeted enrichment, < 0.001% of reads were assigned to parasites. Due to the very small proportion of parasite DNA in our sediment samples (which is other-wise dominated by bacteria and currently unidentifiable DNA, along with plants and fungi), very deep shotgun sequencing would be needed to find parasite DNA in environmental samples. Paleogenetic studies in parasitology are likely to continue to lag behind that seen for other pathogens without changes that make the use of aDNA more cost effective. Microscopy is a relatively cheap and reliable method for egg identification from archeological sediments making it a preferred method for many researchers.

While ancient DNA analysis is a promising addition to the methodological armamentarium in paleoparasitology, our results suggest that currently microscopy remains the most cost-effective and reliable screening tool in paleoparasitology. When resources allow, ELISA and aDNA can be utilized alongside microscopy to analyze a sample more comprehensively. In paleoecological studies as well, sedaDNA has been found to function best as a complementary proxy, as other methods such as palynology and macrofossil identification tend to find similar, but non-overlapping taxa [79–81]. Microscopy continues to successfully recover parasites eggs in samples with low egg concentration. Based on our results comparing eggs per gram concentration generated from microscopic analysis to percent of LCA-assigned reads, there appears to be a minimum concentration of eggs needed for increased chance of DNA recovery using the currently described methods. Samples that contained 30 eggs per gram or lower of *Ascaris* sp. eggs did not contain any detectable *Ascaris* sp. DNA. It is difficult to directly compare this to other studies as many do not report eggs per gram or had higher egg concentrations in all samples studied [13]. However, both in our study and other paleoparasitological studies, there are cases where parasite DNA is detected in the absence of any preserved helminth eggs [14,26,48]. In these cases it is possible that taphonomic changes to the eggs have resulted in inability to identify them by microscopy but that their DNA can still be captured and sequenced. Alternatively, that DNA from adult worms themselves rather than eggs is being detected. It is also likely that eggs are not evenly distributed within a sample and studying different subsamples with microscopy and aDNA will provide different findings.

The complexities of parasite egg preservation and in turn parasite DNA preservation is poorly understood. This diverse group of organisms has variable eggshell structures, rates of embryological development, environmental niches, and are recovered from different contexts including in the gastrointestinal space as in the case of pelvic soil as well as cesspit, latrine, and drain contexts. We were able to detect parasite DNA in various sample types including pelvic soil, coprolites, and latrine or cesspit samples. We had the most success with identifying *Ascaris* sp. using our sedaDNA methods and targeted enrichment. The ability to more easily recover *Ascaris* sp. DNA from all samples may be related to both meth-odological factors as well as parasite biology. As part of its life cycle, *Ascaris* spp. develop in soil, thus by necessity they are resistant to harsh environmental conditions. In addition, extensive reference sequences for *Ascaris* spp. allowed us to create a robust set of baits for *Ascaris* spp. A total of 1,769 of 30,240 baits with average tiling density of 7x were for either

*Ascaris lumbricoides* or *Ascaris suum* (see S2 Appendix for list of baits). In comparison, our bait set had very good coverage of taxa identified by microscopy for which we were not able to recover DNA. *Echinostoma* spp. had a total of 3,213 baits (average tiling density ranged for each species from 8.4–21.8x), *Schistosoma* spp. had 2,262 baits (average tiling density ranged for each species from 8.1–21.8x), and *Dibothriocephalus* spp. (syn. *Diphyllobothrium* spp.) had 1,980 baits (average tiling density ranged for each species from 8.8–23.4x).

Parasite biology will also influence DNA recovery [26]. The size of adult worms from different helminths is variable. For example, *Ascaris lumbricoides* can grow up to 30 cm long whereas *Trichuris trichiura* is typically only 5 cm long and *Dibothriocephalus latus* can grow to 1000 cm [61]. Thus, the biomass of the adult worms would be expected to vary by taxa. This would impact the amount of DNA expected from each species in pelvic soil samples, as pelvic soil samples would contain decomposed intestinal contents including helminths causing infection at the time of death. The number of eggs excreted per day is also quite variable, with *Ascaris lumbricoides* producing around 200,000 eggs per day [82] and *Trichuris trichiura* only 18,000 eggs per day [83]. These factors may be compounded by sedaDNA preservation characteristics, which are thought to necessitate binding to sedimentary minerals [84–87]. If parasite DNA does not come in contact with sedimentary minerals before being broken down by microbes in the local microbiome, those parasite DNA fragments are unlikely to preserve and be recoverable.

While sedaDNA with targeted enrichment has the potential to significantly aid in paleoparasitological research, further methodological improvements are needed to reach the taxonomic characterization already viable with the gold standard method within the field—microscopy. Furthermore, *Giardia duodenalis* antigen was detected using ELISA from four sites, and although our bait set did include protozoa, we were not able to recover any DNA sequences from *Giardia duodenalis*. Thus, assuming that the ELISA signal is real, it does continue to have a unique value in that it is a useful additional method for maximizing recovery of common intestinal protozoa, which are rarely preserved well enough to be visualized using microscopy and have only rarely been detected in aDNA studies thus far [25]. However, the indeterminate results from Must Farm highlight an important limitation of ELISA which is the potential for cross-reactivity with unknown antigens that may be found in soil. Commercial ELISA kits used in this and other studies have very high sensitivity and specificity (>97%) when used on modern stool samples and in their validation there was no cross-reactivity reported for other parasite species tested. However, further exploration of how these performance characteristics change when applied to archaeological samples is needed. Further, optimization of aDNA methods for detection of protozoa would be valuable to confirm ELISA results with a second method.

### Temporal changes in parasites in the Roman period

Using a multimethod approach, we aimed to produce a more accurate reconstruction of parasite infection in the Roman period. Inclusion of pre-Roman and post-Roman samples from areas that were at one time part of the Roman Empire provides important comparative data to understand how parasite diversity may have changed during the Roman period. The samples included in our study contribute a small subset of results for parasite presence in areas that were once part of the Roman Empire from the Neolithic through the medieval period. The parasite taxa recovered from the pre-Roman sites were much more diverse than any other time period. At these sites we recovered various zoonotic parasites including *Echinostoma* sp. and fish tapeworm (*Dibothriocephalus* sp.) which are spread by ingestion of raw or undercooked fish, frogs, or mollusks, as well as *Capillaria* sp. from two sites. This is in conjunction with one common soil-transmitted helminth, whipworm (*Trichuris trichiura*), which is spread by the fecal-oral route.

By comparison in the Roman period, we observe almost exclusively soil-transmitted helminths, roundworm (*Ascaris* sp.) and whipworm (*Trichuris trichiura*). One exception is the site of Arlon, Belgium where *Capillaria* sp. and the lancet liver fluke (*Dicrocoelium dendriticum*) were also recovered. These two taxa of liver flukes are zoonotic parasites that likely represent false parasitism as a result of ingestion of liver from infected animals [88–91]. Though true infections can occur in humans they are rare. Similarly, in medieval period sites, soil-transmitted helminths are almost exclusively present, aside from a coprolite from Jerusalem where fish tapeworm (*Dibothriocephalus* sp.) was also found.

 

When these results are included within the published literature on parasite presence in these time periods, our results are comparable to patterns seen using a much larger dataset (see Tables Fi–Fiii in S1 Appendix for a review of published studies). The proportion of sites with whipworm is consistently high in all time periods, while roundworm appears to increase in frequency in the Roman period and remains stable at that level in the medieval period (Fig 6). Roundworm was recovered from 35% of pre-Roman sites, which increased to 67% of Roman and 79% of medieval period sites. The increase in proportion of sites where roundworm was found in the Roman and medieval periods, and whipworm in the medieval period may reflect decreased sanitation and hygiene. Numerous other factors such as increased urbanization, cultural preferences for the use of chamber pots, and agricultural practices involving the use of human feces as fertilizer may have all contributed to this increase in fecal-oral parasite presence [33,92].

Zoonotic parasites (parasites spread from animals to humans directly or through ingestion of meat) [93], including fish tapeworm (*Dibothriocephalus* sp.), liver flukes (*Dicrocoelium dendriticum* and *Fasciola hepatica*) and beef/pork tapeworm (*Taenia* sp.) are generally more common in pre-Roman time periods compared to the later Roman and medieval periods (Fig 6). This may reflect changes to human-animal interactions and dietary preferences. It is possible that the decline in lancet liver fluke during the Roman period is an anomaly in our data as the proportion of sites where it was found in the medieval period is quite consistent with the pre-Roman period. However, it does appear that *Fasciola* liver fluke was more common in the pre-Roman period compared to later time periods. It was found at 40% of pre-Roman sites compared to 12% of Roman and 12% of medieval sites. Both types of liver flukes in fecal samples can reflect true infection or false parasitism, usually from consumption of animal liver containing eggs [94]. In all time periods, nearly all sites where *Fasciola hepatica* was found are North of the Alps.

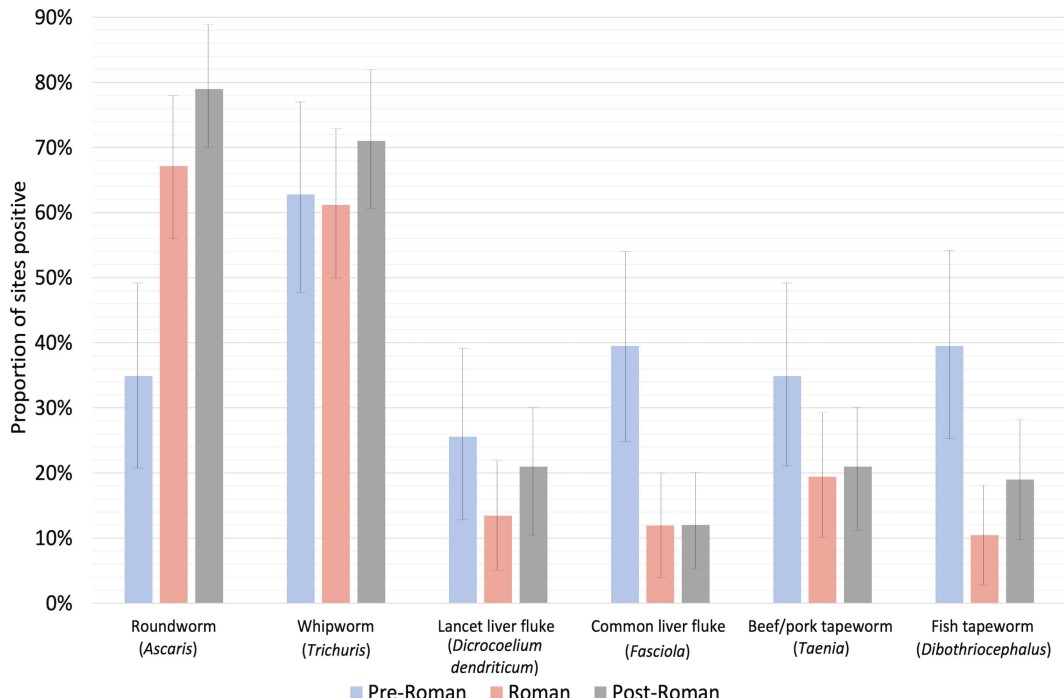

**Fig 6. Temporal trends in parasite diversity before, during, and after the Roman period for the most common helminths recovered from archeological samples.** For each parasite, bars indicate the proportion of studied sites where each parasite was recovered for pre-Roman, Roman, and post-Roman sites with 95% confidence intervals. See S1 Appendix for individual studies used to generate temporal trends.

A similar pattern as seen for liver flukes is seen for fish tapeworm (*Dibothriocephalus* sp.). Fish tapeworm was most common in pre-Roman sites, being found at 40% of sites and then decreases to 10% in the Roman period and 19% in the medieval period. Fish tapeworm is acquired from ingestion of raw or undercooked freshwater fish in Europe. The decline in the Roman period may represent a preference for cooked fish or, alternatively, a decreased reliance on freshwater fish but rather marine fish.

There are many intricacies of transmission and determinants of infection that can be explored for each individual parasite. However, what is most striking overall is the high proportion of sites in all time periods in Europe where we find evidence for enteric parasites. Roundworm is present at 67% of Roman sites studied and whipworm is found at 61% of sites. Of course, a major limitation of this type of analysis is publication bias. However, there is no doubt that these two parasites contributed to the burden of disease in most Roman communities. Many infections with these two species can be asymptomatic; however, they can also cause gastrointestinal upset and for children result in nutrient deficiencies, stunting of growth, and intellectual disability [95].

The vast majority of work done in paleoparasitology or archeoparasitology has relied on microscopic analysis of sediments and paleofeces to identify parasites. ELISA has also been used by some groups to detect ancient protozoa. We have shown that sedaDNA methods combined with targeted enrichment can be used to detect parasite DNA in as little as 0.25 g of paleofecal sediment. A multimethod approach in paleoparasitology allows for the most complete reconstruction of past parasite diversity currently available to us. However, microscopy continues to be the most sensitive method for detecting helminths in paleofecal samples and appears to detect helminth eggs present in lower concentrations than our sedaDNA methods are able to detect. ELISA is uniquely able to detect protozoa such as *Giardia duodenalis* in our samples. In our set of samples, aDNA replicated identification of *Ascaris* sp. presence in samples with decorticated eggs that can be challenging to identify, provided species-level identification in one case, and detected additional parasites not seen by microscopy and ELISA. However, further work optimizing sedaDNA methods for the field of paleoparasitology are needed. As these methods are refined and applied to additional samples, they may help facilitate a more holistic analysis of parasite phylogeography and parasite biology through time.

Applying this multimethod approach to 26 coprolite, latrine, and sewer samples from the Neolithic through medieval period allowed us to characterize temporal trends in parasite infection around the rise, expansion and eventual decline of the Roman Empire. We found a trend of decreased zoonotic parasites from the pre-Roman to Roman and medieval periods and concurrent increase in parasites spread by the fecal-oral route, especially roundworm.

## Supporting information

**S1 Appendix. Materials, methods, and parasite literature review.**
(DOCX)

**S2 Appendix. Parasite bait accession numbers and bait description.**
(XLSX)

**S3 Appendix. Detailed results.**
(XLSX)

## Acknowledgments

We would like to thank local archeological institutions, and excavation teams from the sites included in this study, and the official parties granting the required permissions. We thank Techlab, Blacksburg, USA for their kind donation of ELISA kits used in this research. We thank Christa Clamer for providing material and contextual interpretation of our results. Thanks to Brian Golding for providing access to his computational resources, and to Ana Duggan for assistance and guidance

on bioinformatic workflow. Thanks to all members and affiliates of the McMaster Ancient DNA Centre for their ongoing support of this work. Numerous individuals have contributed to various stages of this work; from the site of Sagalassos we would like to thank the Research Foundation Flanders and the Research Fund of KU Leuven for supporting the research for this paper. From the site of Ephesus we would like to thank the late excavation director Sabine Ladstätter, who made it possible for us to collect samples for analysis. From Vacone we would like to thank Devin Ward, Gary Farney, Dylan Bloy, and the Soprintendenza Archeologia, Belle arti e Paesaggio per l'area metropolitana di Roma e la provincia di Rieti.

## Author contributions

**Conceptualization:** Marissa L. Ledger, Tyler J. Murchie, Piers D. Mitchell, Hendrik Poinar.

**Data curation:** Marissa L. Ledger.

**Formal analysis:** Marissa L. Ledger, Tyler J. Murchie.

**Funding acquisition:** Marissa L. Ledger, Piers D. Mitchell, Hendrik Poinar.

**Investigation:** Marissa L. Ledger, Tyler J. Murchie, Piers D. Mitchell, Hendrik Poinar.

**Methodology:** Marissa L. Ledger, Tyler J. Murchie, Zachery Dickson, Melanie Kuch, Piers D. Mitchell, Hendrik Poinar.

**Project administration:** Melanie Kuch.

**Resources:** Scott D. Haddow, Christopher J. Knüsel, Gil J. Stein, Mike Parker Pearson, Rachel Ballantyne, Mark Knight, Koen Deforce, Maureen Carroll, Candace Rice, Tyler Franconi, Nataša Šarkić, Saša Redžič, Erica Rowan, Nicholas Cahill, Jeroen Poblome, Maria de Fátima Palma, Helmut Brückner, Hendrik Poinar.

**Supervision:** Piers D. Mitchell, Hendrik Poinar.

**Validation:** Marissa L. Ledger.

**Visualization:** Marissa L. Ledger.

**Writing – original draft:** Marissa L. Ledger, Tyler J. Murchie, Zachery Dickson, Piers D. Mitchell, Hendrik Poinar.

**Writing – review & editing:** Marissa L. Ledger, Tyler J. Murchie, Zachery Dickson, Scott D. Haddow, Christopher J. Knüsel, Gil J. Stein, Mike Parker Pearson, Rachel Ballantyne, Mark Knight, Koen Deforce, Maureen Carroll, Candace Rice, Tyler Franconi, Nataša Šarkić, Erica Rowan, Nicholas Cahill, Jeroen Poblome, Maria de Fátima Palma, Helmut Brückner, Piers D. Mitchell, Hendrik Poinar.

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
