## [Decision Letter · Decision Letter 0]

4 Dec 2024

Response to Reviewers
Revised Manuscript with Track Changes
Manuscript

Shaden Kamhawi

co-Editor-in-Chief

Paul Brindley

co-Editor-in-Chief

**Journal Requirements:**

1) We do not publish any copyright or trademark symbols that usually accompany proprietary names, eg ©,  ®, or TM  (e.g. next to drug or reagent names). Therefore please remove all instances of trademark/copyright symbols throughout the text, including:

- © on page: 26

- TM on page: 7.

3) Some material included in your submission may be copyrighted. According to PLOSu2019s copyright policy, authors who use figures or other material (e.g., graphics, clipart, maps) from another author or copyright holder must demonstrate or obtain permission to publish this material under the Creative Commons Attribution 4.0 International (CC BY 4.0) License used by PLOS journals. Please closely review the details of PLOSu2019s copyright requirements here: PLOS Licenses and Copyright. If you need to request permissions from a copyright holder, you may use PLOS's Copyright Content Permission form.

Potential Copyright Issues:

i) Figure 1. Please (a) provide a direct link to the base layer of the map (i.e., the country or region border shape) and ensure this is also included in the figure legend; and (b) provide a link to the terms of use / license information for the base layer image or shapefile. We cannot publish proprietary or copyrighted maps (e.g. Google Maps, Mapquest) and the terms of use for your map base layer must be compatible with our CC BY 4.0 license.

4) Thank you for providing your Data Availability Statement. Please provide us with a direct link to access the dataset. If your manuscript is accepted for publication, you will be asked to provide these details on a very short timeline. We therefore suggest that you provide this information now, though we will not hold up the peer review process if you are unable.

2) State what role the funders took in the study. If the funders had no role in your study, please state: "The funders had no role in study design, data collection and analysis, decision to publish, or preparation of the manuscript.".

If you did not receive any funding for this study, please simply state: The authors received no specific funding for this work.

6)  Please ensure that the funders and grant numbers match between the Financial Disclosure field and the Funding Information tab in your submission form. Note that the funders must be provided in the same order in both places as well.  Currently, this funding information "Tidmarsh Cambridge Scholarship from the Cambridge Commonwealth and TJM and HNP were funded by the CANA Foundation, the Garfield Weston Foundation, the Social Sciences and Humanities Research Council of Canada, CIFAR" is missing from the Funding Information tab.                                                . 

Please indicate by return email the full and correct funding information for your study and confirm the order in which funding contributions should appear. Please be sure to indicate whether the funders played any role in the study design, data collection and analysis, decision to publish, or preparation of the manuscript.

**Comments to the author:**

Please note that one of the reviews is uploaded as an attachment.

**Reviewers' comments:**

**Key Review Criteria Required for Acceptance?**

**Methods**

-Are the objectives of the study clearly articulated with a clear testable hypothesis stated?

-Is the study design appropriate to address the stated objectives?

-Is the population clearly described and appropriate for the hypothesis being tested?

-Is the sample size sufficient to ensure adequate power to address the hypothesis being tested?

-Were correct statistical analysis used to support conclusions?

-Are there concerns about ethical or regulatory requirements being met?

Reviewer #1: This work is well designed and followed valuable points of views to illustrate results to the readers. Those who are interested to work on paleoparasitology will encourage to expand their program in to all three possibilities of identifying parasites of ancient times.

As the last word I feel well satisfied by this manuscript. I would like to congratulate the organizers of this very good team work

Reviewer #2: -Are the objectives of the study clearly articulated with a clear testable hypothesis stated? Yes

-Is the study design appropriate to address the stated objectives? Yes

-Is the population clearly described and appropriate for the hypothesis being tested? Yes

-Is the sample size sufficient to ensure adequate power to address the hypothesis being tested?

There is a very small sample size in this study, however, this tends to be the case when working with ancient DNA. The limitation is clearly recognized in the study.

-Were correct statistical analysis used to support conclusions? Yes

-Are there concerns about ethical or regulatory requirements being met? No

Reviewer #3: (No Response)

**Results**

-Does the analysis presented match the analysis plan?

-Are the results clearly and completely presented?

-Are the figures (Tables, Images) of sufficient quality for clarity?

Reviewer #1: (No Response)

Reviewer #2: -Does the analysis presented match the analysis plan? Yes

-Are the results clearly and completely presented? Yes

-Are the figures (Tables, Images) of sufficient quality for clarity? Yes. Nevertheless, data from supplementary figures F1 to F3 and Figure 2 could have been combined in a presence/absence heatmap, showing the congruence between this study's findings and that of published results.

Reviewer #3: (No Response)

**Conclusions**

-Are the conclusions supported by the data presented?

-Are the limitations of analysis clearly described?

-Do the authors discuss how these data can be helpful to advance our understanding of the topic under study?

-Is public health relevance addressed?

Reviewer #1: (No Response)

Reviewer #2: -Are the conclusions supported by the data presented? Yes

-Are the limitations of analysis clearly described? The authors highlight the constraints of their limited sample size and make grounded conclusions accordingly.

-Do the authors discuss how these data can be helpful to advance our understanding of the topic under study? Yes

-Is public health relevance addressed? Yes

Reviewer #3: (No Response)

**Editorial and Data Presentation Modifications?**

Reviewer #1: (No Response)

Reviewer #2: An additional figure in the article illustrating the presence or absence of parasites in previous studies alongside this study's findings would have been helpful to the reader, highlighting the congruence of results.

However, this is just a suggestion, and it does not condition the suitability of the article.

Reviewer #3: (No Response)

**Summary and General Comments**

Reviewer #1: (No Response)

Reviewer #2: The article looks into the parasitic load of pre-Roman, Roman and post-Roman populations in Europe through the analysis of coprolites. Using a combination of microscopy, ELISA and (antigens) and aDNA shotgun sequencing the study compares each method and recommends a combination of all three for optimal parasitic diagnostics.

The article is very well structured and easy to follow in a very detailed and concise manner. The discussion is particularly insightful and does not overreach conclusionwise. However, ELISA is very briefly discussed in the paper despite it being recognized as one of the three pilar methods for parasite diagnostics. The lack of discussion of ELISA findings and recognizing a bias/contamination in some ELISA results, despite no explanation or correction to this given in the paper or the appendix, makes ELISA substantially less convincing than microscopy and aDNA.

The analysis was straight to the point. From the appendix it is also clear that very thorough research was done on published Paleoparasitological studies, providing records of parasites found in these studies throughout time. The study could have added this information to a figure in the same format as Figure 2, but presence or abscence of a parasite as opposed to quantitative analyses, including the findings of previous works, giving a more in-depth geospatial representation of parasitic loads through time from this study and previous studies.

Furthermore, the authors have declared the data will be publicly available. This has not been done yet, but is expected shortly after submission.

Reviewer #3: (the pdf with the correction is in the attachment)

The article provides a clear, concise, and relevant study on paleoparasitology and paleogenomics. The authors successfully develop a context that is well-researched and coherent, establishing the objectives of the study. However, there is information that is basically part of the results and discussion on the topic of Methodology, as well as phrases of discussion on the topic of Results; I marked some of these phrases in the pdf file. Regrettably, it is stated in the introduction that the paleogenetic analysis of parasites has been little addressed in the past, the authors ignore an extensive contribution of studies mainly in archaeological and historical sites, especially in Brazil, but also Chile and Argentina. The taxonomic writing should be corrected along with the manuscript; I also made some corrections in the pdf file.

I have only two main worries. 1) How the species Ascaris lumbricoides and A. suum were identified since these two parasites are not genetically separated using most of the molecular markers? 2) The identification of Ascaris spp. from atypical morphology eggs is controversial since the DNA was not extracted specifically from these atypical eggs but from the sediment that could be, in fact, positive to the parasite.

PLOS authors have the option to publish the peer review history of their article (what does this mean? ). If published, this will include your full peer review and any attached files.

**Do you want your identity to be public for this peer review?** For information about this choice, including consent withdrawal, please see our Privacy Policy .

Reviewer #1: **Yes: ** Mowlavi Gholamreza School of Public Health Tehran University of Medical Sciences

Reviewer #2: **Yes: ** Arve Lee Willingham Grijalba

Reviewer #3: **Yes: ** Alena Iñiguez

**Figure resubmission:****Reproducibility:** To enhance the reproducibility of your results, we recommend that authors of applicable studies deposit laboratory protocols in protocols.io, where a protocol can be assigned its own identifier (DOI) such that it can be cited independently in the future. Additionally, PLOS ONE offers an option to publish peer-reviewed clinical study protocols. Read more information on sharing protocols at https://plos.org/protocols?utm_medium=editorial-email&utm_source=authorletters&utm_campaign=protocols 

---

## [Decision Letter · Decision Letter 1]

10 Mar 2025

Sedimentary ancient DNA as part of a multimethod paleoparasitology approach reveals temporal trends in human parasitic burden in the Roman period

Dear Dr. Ledger,

Thank you for submitting your manuscript to PLOS Neglected Tropical Diseases. After careful consideration, we feel that it has merit but does not fully meet PLOS Neglected Tropical Diseases's publication criteria as it currently stands. Therefore, we invite you to submit a revised version of the manuscript that addresses the points raised during the review process.

Please submit your revised manuscript within 60 days Apr 09 2025 11:59PM. If you will need more time than this to complete your revisions, please reply to this message or contact the journal office at plosntds@plos.org. Please include the following items when submitting your revised manuscript:

We look forward to receiving your revised manuscript.

Kind regards,

Elham Kazemirad, Ph.D

Academic Editor

Krystyna Cwiklinski

Section Editor

Shaden Kamhawi

co-Editor-in-Chief

Paul Brindley

co-Editor-in-Chief

**Additional Editor Comments:**

Please prepare the manuscript according to the PLOS style. As the reviewer requested, the data should be available in NCBI and ENA - I am assuming the accession number provided is embargoed until publication?

**Reviewers' Comments:**

Reviewer's Responses to Questions

**Key Review Criteria Required for Acceptance?**

**Methods:**

-Are the objectives of the study clearly articulated with a clear testable hypothesis stated?

-Is the study design appropriate to address the stated objectives?

-Is the population clearly described and appropriate for the hypothesis being tested?

-Is the sample size sufficient to ensure adequate power to address the hypothesis being tested?

-Were correct statistical analysis used to support conclusions?

-Are there concerns about ethical or regulatory requirements being met?

Reviewer #2: I

Reviewer #3: The article was modified according to the suggestions made.

**Results:**

-Does the analysis presented match the analysis plan?

-Are the results clearly and completely presented?

-Are the figures (Tables, Images) of sufficient quality for clarity?

Reviewer #2: (No Response)

Reviewer #3: The article was modified according to the suggestions made.

Some taxonomic names require correction before publication. Please review the revised PDF.

**Conclusions:**

-Are the conclusions supported by the data presented?

-Are the limitations of analysis clearly described?

-Do the authors discuss how these data can be helpful to advance our understanding of the topic under study?

-Is public health relevance addressed?

Reviewer #2: (No Response)

Reviewer #3: The article was modified according to the suggestions made.

Some taxonomic names require correction before publication. Please review the revised PDF.

**Editorial and Data Presentation Modifications?**

Reviewer #2: (No Response)

Reviewer #3: A “Minor Revision” is suggested to address the incorrect taxonomic writing that remains.

**Summary and General Comments:**

Reviewer #2: I will stick to all of my previous comments. The article is very well written and easy to follow. Thank you to the authors for responding to my comments. Nevertheless, I have searched for the public data again and have still not found it in both NCBI and ENA.

Reviewer #3: The article was modified according to the suggestions made.

PLOS authors have the option to publish the peer review history of their article (what does this mean? ). If published, this will include your full peer review and any attached files.

**Do you want your identity to be public for this peer review?** For information about this choice, including consent withdrawal, please see our Privacy Policy .

Reviewer #2: **Yes: ** Arve Lee Willingham Grijalba

Reviewer #3: **Yes: ** Alena M. Iñiguez

**Figure resubmission:**

**Reproducibility:**



---

## [Decision Letter · Decision Letter 2]

12 May 2025

Dear Dr Ledger,

We are pleased to inform you that your manuscript 'Sedimentary ancient DNA as part of a multimethod paleoparasitology approach reveals temporal trends in human parasitic burden in the Roman period' has been provisionally accepted for publication in PLOS Neglected Tropical Diseases.

Best regards,

Elham Kazemirad, Ph.D

Academic Editor

Krystyna Cwiklinski

Section Editor

Shaden Kamhawi

co-Editor-in-Chief

Paul Brindley

co-Editor-in-Chief

**Please check and edit sp (singular) or spp (plural) for *Ascaris* and other genus in all the text.**

Reviewer's Responses to Questions

**Key Review Criteria Required for Acceptance?**

**Methods**

-Are the objectives of the study clearly articulated with a clear testable hypothesis stated?

-Is the study design appropriate to address the stated objectives?

-Is the population clearly described and appropriate for the hypothesis being tested?

-Is the sample size sufficient to ensure adequate power to address the hypothesis being tested?

-Were correct statistical analysis used to support conclusions?

-Are there concerns about ethical or regulatory requirements being met?

Reviewer #3: ok

**Results**

-Does the analysis presented match the analysis plan?

-Are the results clearly and completely presented?

-Are the figures (Tables, Images) of sufficient quality for clarity?

Reviewer #3: ok

**Conclusions**

-Are the conclusions supported by the data presented?

-Are the limitations of analysis clearly described?

-Do the authors discuss how these data can be helpful to advance our understanding of the topic under study?

-Is public health relevance addressed?

Reviewer #3: ok

**Editorial and Data Presentation Modifications?**

Reviewer #3: Accept

**Summary and General Comments**

Reviewer #3: The authors corrected the taxonomic writing issues.

PLOS authors have the option to publish the peer review history of their article (what does this mean? ). If published, this will include your full peer review and any attached files.

**Do you want your identity to be public for this peer review?** For information about this choice, including consent withdrawal, please see our Privacy Policy .

Reviewer #3: **Yes: ** Alena Iñiguez

---

## [Editor Report · Acceptance letter]

Dear Dr Ledger,

We are delighted to inform you that your manuscript, "Sedimentary ancient DNA as part of a multimethod paleoparasitology approach reveals temporal trends in human parasitic burden in the Roman period," has been formally accepted for publication in PLOS Neglected Tropical Diseases.

Best regards,

Shaden Kamhawi

co-Editor-in-Chief

Paul Brindley

co-Editor-in-Chief
